# Counterfactual Optimization of Treatment Policies Based on Temporal Point Processes

Zilin Jing [1]  Chao Yang [1]  Shuang Li [1]

## Abstract

In high-stakes areas such as healthcare, it is interesting to ask counterfactual questions: what if some executed treatments had been performed earlier/later or changed to other types? Answering such questions can help us debug the observational treatment policies and further improve the treatment strategy. Existing methods mainly focus on generating the whole counterfactual trajectory, which provides overwhelming information and lacks specific feedback on improving certain actions. In this paper, we propose a counterfactual treatment optimization framework where we optimize specific treatment actions by sampling counterfactual symptom rollouts and meanwhile satisfying medical rule constraints. Our method can not only help people debug their specific treatments but also has strong robustness when training data are limited.

## 1. Introduction

Although deep reinforcement learning policies have been investigated for sepsis patients in ICU to aid the treatment strategy design and have achieved promising results (Komorowski et al., 2018), the *lack of interpretability* of the learned *black-box* policies hinders their wide applications in real life. Clinicians are not satisfied with knowing which action to take but are also interested in understanding why to take such actions and how to improve the treatment actions.

In this paper, we focus on answering the following *what-if* question: *given the observational treatment and outcome trajectories, could we perturb specific treatment actions so that the outcome is optimized in a counterfactual manner?* Meanwhile, the perturbations should at least satisfy or not violate some medical rules. For example, to mitigate the an-

tibiotic's persistence, different antibiotics should be applied subject to reasonable temporal orders. For another example, when a person aims to develop a healthy exercise habit, it is inappropriate to recommend him/her excessively exercising and drastically reducing food intake. Such recommendations are unsustainable and should be avoided.

Recently, a counterfactual off-policy evaluation method has been developed for partially observable Markov Decision Process (POMDP) (Oberst & Sontag, 2019) and has been extended to multivariate Hawkes processes (Noorbakhsh & Rodriguez, 2022). This paper aims to move one step further and propose to optimize the treatment policies by counterfactually perturbing the observational actions. Especially we assume that the occurrence of symptoms of an individual is networked or intertwined and can be modeled as a multivariate temporal point process. The optimal perturbation of the actions is obtained based on the counterfactual sampling of the symptom rollouts.

**Motivating Example** Novice doctors may give suboptimal policies to their patients. Given observational treatment and symptom traces, we adopt the Gumbel-max trick to sample counterfactual symptom sequences by perturbing the treatment actions in terms of time and types; the resulting outcome is evaluated retrospectively. The results will provide quantitative insight into how to further improve the treatment actions.

## 2. Related Work

**Counterfactual Sampling** The literature on counterfactual sampling mainly focuses on using Structural Causal Model (SCM) in Markov Decision Process and generating counterfactual sequences (Oberst & Sontag, 2019)(Tsirtsis et al., 2021). These models enable doctors to debug their policies by comparing observed and counterfactual sequences. However, most of them are limited to discrete-time settings and ignore the influence of historical trajectories. On the other hand, (Hızlı et al., 2022) and (Schulam & Saria, 2017) use a mixture of Gaussian processes to monitor doctors' treatment and patients' outcomes in continuous time. (Seedat et al., 2022) incorporates neural controlled differential equations to deal with irregular samples. Two most recent papers (Noorbakhsh & Rodriguez, 2022) and (Hızlı et al., 2022)

---

[1]School of Data Science, The Chinese University of Hong Kong (Shenzhen), China . Correspondence to: Shuang Li <lishuang@cuhk.edu.cn>.

*Workshop on Interpretable ML in Healthcare at International Conference on Machine Learning (ICML)*, Honolulu, Hawaii, USA. 2023. Copyright 2023 by the author(s).

integrate SCM into the temporal point process and sample counterfactual sequences in the past. However, these papers still focus on generating counterfactual samples given small permutations on the past trajectory. In contrast, we explore how to apply counterfactual rollouts to optimize treatment policies and also incorporate some temporal logic rules to exclude unreasonable trajectories.

## 3. Preliminaries

**Temporal Point Process (TPP)** TPP is a stochastic process that models a sequence of discrete events localized in continuous time (Rasmussen, 2018). Define a counting process $N(t)$, which records how many events occur before time $t$. The TPP is characterized by the happening rate of an event, i.e., intensity function denoted as $\lambda(t) := \mathbb{E}[dN(t)]/dt$, where $\mathbb{E}[dN(t)]$ denotes the number of events happening in $N([t, t + dt])$.

If each event has a specific event type, we name it Marked Temporal Point Process (MTPP), where the intensity is a function of time and event type. The joint conditional intensity function is

$$\lambda(t, k) = \lambda(t) * f(k \mid t)$$

where $k$ is the marker of the event and $f(k \mid t)$ is the *conditional probability mass function* (given discrete marker) of event type $k$ given $t$. Given an analytical expression of intensity function $\lambda(t)$ and under some mild conditions, we can easily sample events from it with Ogata thinning algorithm (see Appendix A).

Multivariate Hawkes Prcoess is a common historical dependent MTPP. Suppose $H_{t_N} = \{(t_i, k_i) : t_i < t, 1 \leq k_i \leq M, 1 \leq i \leq N\}$ is a multivariate Hawkes Prcoess up to time $t$ with $M$ mark dimensions and $N$ events. The conditional intensity function of $m$ th mark dimension is defined as

$$\lambda_m(t|H_{t_N}) = \mu_m + \sum_{\{k:1 \leq k \leq M\}} \sum_{\{i:t_i < t, k_i = k\}} \alpha_{mk} e^{-\beta_{mk}(t - t_i)}$$

Recently, neural temporal point process has been widely used to predict the time and mark of next event. Compared with MTPP, it is more flexible and efficient. For instance, (Zuo et al., 2020) represents the conditional intensity function as

$$\lambda_m(t|H_{t_N}) = f_m(\alpha_m \frac{t - t_N}{t_N} + w_m^\top h(H_{t_N}) + b_m)$$

where $h(H_{t_N})$ is the encoding of historical events.

**Temporal Logic Rule** can be used to define medical rules that the counterfactual sampling should preserve. Specifically, we introduce *predicate* as a logic variable, which is a Boolean function $X(\cdot)$ that is defined over a set of entities $\mathcal{C}$, such as

$$X(\cdot) : C_1 \times \ldots C_N \to \{0, 1\}$$

where $X(\cdot)$ can be either properties of an entity (e.g., Diabetes($C$): whether C has diabetes) or relations of entities (e.g., Sibling($C_1, C_2$): whether $C_1$ and $C_2$ are siblings). The medical rule can be expressed as a compact set of IF-THEN logic rules, such as

$$f_1 : \text{Diabetes}(X) \wedge \text{Sibling}(X, Y) \to \text{Diabetes}(Y).$$

Further, we introduce temporal order constraints as Boolean predicates to the logic rules, e.g., Before(Drug1, Drug2)= $1\{t_2 - t_1 > 0\}$ which is true whenever drug1 is applied earlier than drug2. The IF-THEN rules can be converted to the conjunctive normal form (CNF), such as

$$f_2 : \neg\text{Diabetes}(X) \vee \neg\text{Sibling}(X, Y) \vee \text{Diabetes}(Y).$$

which is a Boolean formulas expressed as conjunctions of clauses with an AND or OR. We will use the CNF expression to construct the *regularization term* in the counterfactual optimization objective functions. Here, we will treat the medical rules as *soft constraints*. Inspired by Zhou et al. (2021), we construct penalties from temporal logic rules to exclude unreasonable counterfactual treatment in our optimization process. We extend the predicate from Boolean value 0/1 to continuous value in $[0, 1]$ and convert the logical operators to arithmetic operators, such as $x_1 \wedge x_2 = \max\{x_1 + x_2 - 1, 0\}, x_1 \vee x_2 = \min\{x_1 + x_2, 1\}$, and $\neg x = 1 - x$. In this way, if any logic rule is not satisfied, we can define the distance to satisfaction as

$$\max\left\{1 - \sum_{i \in I_j^+} y_i - \sum_{i \in I_j^-} (1 - y_j), 0\right\} \quad (1)$$

where $I_j^+$ refers to sets of predicates that should attain value 1 in the logic rule, and $I_j^-$ refers to sets of predicates that should attain value 0 in the logic rule.

**Structural Causal Model (SCM)** The counterfactual analysis requires the knowledge of structural casual model (SCM) (Pearl, 2009), denoted as $\mathcal{M}$. An SCM is a triple $\mathcal{M} = (U, X, F)$, which consists of a set of unexplained exogenous "noise" variables $U$, a set of endogeneous variables $X$, and a set of deterministic functions $F = \{f_1, f_2, ..., f_n\}$ of the form $X_i = f_i(PA_i, U_i)$, where $PA_i \subseteq X \setminus X_i$. We assume that no unmeasured confounders and $U$ are jointly independent. SCM can be used to present how the outcome $Y$ is caused by the treatment $T$ given the confounding variables. Given the known SCM model $\mathcal{M}$, we can compute the *conditional average treatment effect* (CATE) using the *do-operator*, i.e., $\tau_x = E[Y|X, do(T = 1)] - E[Y|X, do(T = 0)]$ where $X$ is the confounding variables. To further estimate the *counterfactual effect* $\tau'$, we

need first infer the posterior latent variables $U$ under present treatment and outcomes, e.g.,

$$P(U = u | X = x, Y = y, T = i)$$

Then we execute $do(T = j)$ and pass the posterior distributions through the modified SCM to infer the counterfactual outcome.

**Counterfactual Temporal Point Process**  Specific to TPPs, Noorbakhsh & Rodriguez (2022) proposes a counterfactual sampling framework, which integrates the counterfactual off-policy evaluation methods for SCM with the Lewis thinning algorithm. Suppose a TPP has intensity $\lambda(t) \leq \lambda_{max}$, one can first sample a sequence of potential events by sampling intervals $\Delta t = \frac{-\ln u}{\lambda_{max}}$, where $u \sim U(0,1)$. Then for each point, we construct a SCM with a binary final outcome $Y$, indicating the acceptance ($Y = 1$) or rejection ($Y = 0$) of the point. The outcome $Y$ has the following structural equation:

$$Y_i = \underset{y \in \{0,1\}}{\arg\max} \log p(Y_i = y | \lambda(t_i)) + U_i \quad (2)$$

with $p(Y_i = y | \lambda(t_i)) = y \cdot p(\lambda(t_i)) + (1-y) \cdot (1 - p(\lambda(t_i)))$, $p(\lambda(t_i)) = \lambda(t_i)/\lambda_{max}$, $U_i \sim \text{Gumbel}(0,1)$, and $t_i \sim \lambda_{max}$. Given a sequence of observed accepted events $\mathcal{H}_m$ and rejected events $\mathcal{H}_{max} \setminus \mathcal{H}_m$ given intensity $\lambda_m(t)$ determined by the observed events. One can first estimate the posterior noise distributions and then get a Monte-Carlo estimate of the counterfactual thinning probability, denoted as $p^{do(T_i = \lambda_{m'}(t_i))}(Y_i = y)$ as

$$\mathbb{E}_{U_i | Y_i = y_i, \lambda_m(t_i)} \left[ \mathbb{I}[y = \underset{y' \in \{0,1\}}{\arg\max} \log p(Y_i = y' \mid \lambda_{m'}(t_i)) + U_i] \right]$$

Here $do(T_i = \lambda_{m'}(t_i)))$ equals to assigning the conditional intensity function $\lambda_{m'}(t)$ at time $t_i$.

## 4. Model

Let's consider a setting similar to the dynamic treatment regime (DTR), where the total number of treatments is fixed (See Fig.1). Define Treatment event as $A_{1:n} = \{(t_i, s_i)\}_{i=1,\ldots n}$, with treatment types $\{s_i\}_{i=1,\ldots,n}$ are observed at time $\{t_i\}_{i=1,\ldots n}$. Similarly, Symptom events are $X_{1:m} = \{(t_j, s_j)\}_{j=1,\ldots m}$, with symptom types $s_j$. Both events are modeled as multivariate TPPs and each intensity function depends on historical events. We will introduce flexible TPP models (Du et al., 2016; Zuo et al., 2020) to estimate the intensity functions from batch data. The final outcome $Y$ refers to the survival time.

We aim to answer what-if questions: Given the observed treatment and symptom traces, how could we perturb the treatment actions $do(A_{1:n} = a'_{1:n})$ so that the survival time is optimized in a counterfactual manner? Suppose original action is $a_i = (t_i, s_i)$, we could perturb it in three ways:

1) Fix action type and only change its action time: $do(A_i = (s_i, t_i + \Delta t))$.

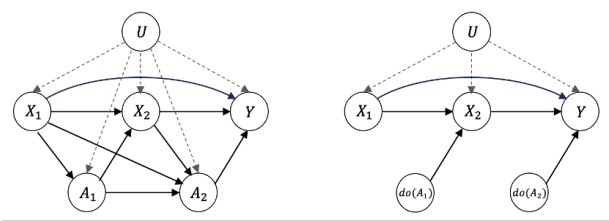

*Figure 1.* Observed causal diagram (left) and counterfactual diagram (right) of DTR with two actions

2) Fix action time and only change its action type: $do(A_i = (s'_i, t_i))$.

3) Jointly change action time and action type: $do(A_i = (s'_i, t_i + \Delta t))$.

In addition, to make the counterfactual changes as close as real-life setting, we add some temporal logic rules as constraints to penalize any unreasonable counterfactual treatments. For instance, we cannot give too aggressive treatment consecutively to older people, which may cause great damage to their bodies. Suppose there are $K$ logic rules, our goal is to maximize the logic-regularized reward function:

$$\mathcal{R}(a'_{1:n}) := \mathbb{E}[Y | do(A_{1:n} = a'_{1:n})] - \sum_{k=1}^{K} w_k \phi_k(a'_{1:n}) \quad (3)$$

where $\phi_k(a'_{1:n})$ can be computed as in Eq. (1).

## 5. Counterfactual Optimization

In this section, we propose a counterfactual optimization algorithm to optimize target function $\mathcal{R}(a'_{1:n})$. Suppose counterfactual action is parameterized by policy $\pi_\theta$ We first propose a counterfactual sampling algorithm to calculate survival time $Y$ under the counterfactual policy. Then we calculate the reward of counterfactual rollouts to optimize treatment policies.

### 5.1. Counterfactual Sampling

In Alg.1, we input an observed sequence $H$, a user-defined model $Z$, the time limit $T$ and an initialized policy $\pi_\theta$. Model $Z$ takes an observed point process $H$ as input and outputs future conditional intensity function $\lambda(t, s)$. For each event $(t_i, s_i)$ in the observed sequence $H$, if $s_i$ is an action event, we sample a counterfactual action from $\pi_\theta$. If $s_i$ is an outcome event, we resample its acceptance decision using Gumbel-max sampling algorithm.

In our paper, we use transformer hawkes process (Zuo et al., 2020) and define decaying kernel for output condition intensity function, so both $\lambda_{obs}(t, s)$ and $\lambda_{cf}(t, s)$ are bounded at each starting point. For sequence $H_{t_N} = \{(t_j, s_j)\}_{j=1}^{N}$, we separately encode its temporal information $t_j$ and event mark $s_j$. Then we concatenate them together and pass them through multiple encoder layers to get embeddings $h(H_{t_j})$ at time $t_j, j = 1, 2 \ldots N$.

Suppose the present time is $t$ and last event happens at $t_j$, the conditional intensity function for type $k$ is

$$\lambda_k(t|H_t) = f_k(\alpha\frac{t-t_j}{t_j} + w_k^\top h(H_{t_j}) + b_k)$$

where $f_k(x) = \beta \cdot \log(1 + \exp(\frac{x}{\beta}))$ is a smooth function.

We can use intensity function to predict next event time:

$$p(t|H_t) = \lambda(t|H_t)\exp(-\int_{t_j}^{t} \lambda(\tau|H_\tau)d\tau)$$

$$\hat{t}_{j+1} = \int_{t_j}^{\infty} t \cdot p(t|H_t)dt$$

Suppose the mark only depends on historical events. If it is discrete, we can monitor it with a multinomial distribution

$$p(s_{j+1}^k|H_t) = \frac{\exp(V_{j+1}^k h(H_{t_j}) + b_{j+1}^k)}{\sum_{k=1}^{M} \exp(V_{j+1}^k h(H_{t_j}) + b_{j+1}^k)}$$

$$\hat{s}_{j+1} = \underset{k}{\operatorname{argmax}}\frac{p(s_{j+1}^k|H_{t_j})}{\sum_{k=1}^{M} p(s_{j+1}^k|H_{t_j})}$$

The likelihood of the sequence $H_{t_N}$ is

$$l(H_{t_N}) = \sum_{j=1}^{N} \log\lambda(t_j, s_j|H_{t_j}) - \int_{t_1}^{t_N} \lambda(t|H_t)dt$$

Then at each event of sequence, we could predict the time and mark of next event and compute the BCE loss of mark $L_{s_j}$ and Root Mean Square Error (RMSE) of time $L_{t_j}$

The total loss function to minize consists of negative log likelihood and sum of time prediction loss and mark prediction loss:

$$-l(H_{t_N}) + \sum_{j=2}^{N}(L_{s_j} + L_{t_j}) \tag{4}$$

Note that the prediction loss starting from the second event.

### 5.2. Optimization with Counterfactual Sequences

Define $\pi_{\theta_t}$ and $\pi_{\theta_a}$ as counterfactual policies of changing action time and action type. To optimize the two policies, we use REINFORCE algorithm and continuously sample rollouts (see Appendix C). For each iteration, we sample $n$ counterfactual trajectories to estimate the empirical value of gradient $\theta$. Each trajectory can be sampled by Alg.1. We then calculate survival time and generate rewards to update our policy. The action type is discrete and we parameterize with softmax function

$$\pi(a_i = u) = \frac{\exp(\theta_u)}{\sum_{u'=1}^{U} \exp(\theta_{u'})} \tag{5}$$

where $u$ is the potential treatment index. The action time is continuous and we parameterize with Gaussian kernel with learnable parameters $(\mu_i, \sigma_i)$.

$$\pi(\Delta t_i) = \frac{1}{\sigma_i\sqrt{2\pi}}\exp\left(-\frac{(\mu_i - \Delta t_i)^2}{2\sigma_i^2}\right) \tag{6}$$

---

**Algorithm 1** Counterfactual Sampling Algorithm

---

**Input** : $T, \lambda_{obs}(t,s), \lambda_{cf}(t,s), \mathcal{H} = \{(t_1, s_1), ...,$
$\qquad (t_n, s_n)\}$, Dynamic model $Z$, Policy $\pi(\theta)$
**Initialize :** $t = 0, H' = \emptyset$

1 **while** $t < T$ **do**
2 $\quad \lambda_{max} = \underset{s\in(t,t+l(t))}{max}(\lambda_{obs}(t,s), \lambda_{cf}(t,s))$
3 $\quad \Delta t = \frac{-\log u}{\lambda_{max}}$
4 $\quad r_0 := (t_0, s_0) = H[0]$
5 $\quad$ **if** $s_0$ is action type **then**
6 $\quad\quad$ Take action $do(a_{t',s'}) \in \pi(\theta)$
$\quad\quad H' = H' \cup (t', s')$
$\quad\quad t = t'$
$\quad\quad$ H.pop(0)
7 $\quad$ **end**
8 $\quad$ **else**
9 $\quad\quad$ **if** $t_0 \in (t, t+\Delta t)$ **then**
10 $\quad\quad\quad$ u = Gumbelmax_sampling($\lambda_{obs}(t_0, s),$
$\quad\quad\quad \lambda_{cf}(t_0, s), \lambda_{max}, t_0, H)$
$\quad\quad\quad t = t_0$
$\quad\quad\quad (t_0, s_0) = $ H.pop(0)
$\quad\quad\quad$ **if** $u == 1$ **then**
11 $\quad\quad\quad\quad$ Draw event type $s_i == $ Counterfactual
$\quad\quad\quad\quad$ Mark $(\lambda_{obs(t_0, s_i)}, \lambda_{cf}(t_0, s_i), s_0$
$\quad\quad\quad\quad H' = H' \cup (t_0, s_i)$
12 $\quad\quad\quad$ **end**
13 $\quad\quad$ **end**
14 $\quad\quad$ **else**
15 $\quad\quad\quad$ u = Gumbelmax_sampling($\lambda_{obs}(t, s),$
$\quad\quad\quad \lambda_{cf}(t, s), \lambda_{max}, t+\Delta t, H)$
$\quad\quad\quad t = t + \Delta t$
$\quad\quad\quad$ **if** $u == 1$ **then**
16 $\quad\quad\quad\quad$ Draw event type $s_i \sim \frac{\lambda_{cf}(t_0, s_i)}{\sum_i \lambda_{cf}(t_0, s_i)}$
$\quad\quad\quad\quad H' = H' \cup (t_0, s_i)$
17 $\quad\quad\quad$ **end**
18 $\quad\quad$ **end**
19 $\quad$ **end**
20 $\quad$ Update $\lambda_{cf}(t, s) = $ Dynamic model $(H')$
21 **end**
22 **return** $H'$

---

## 6. Experiment on MIMIC Data

**Experiment setup** MIMIC-IV is an electronic health record dataset, including patients in ICU systems. We validate our counterfactual optimization algorithm on sepsis patients from MIMIC IV. We extracte four lab measurements, including Blood pressure (BP), Arterial Base Excess (ArterialBE), Creatinine, and Urine. Each lab measurements we consider three types of markers (low, normal, high).

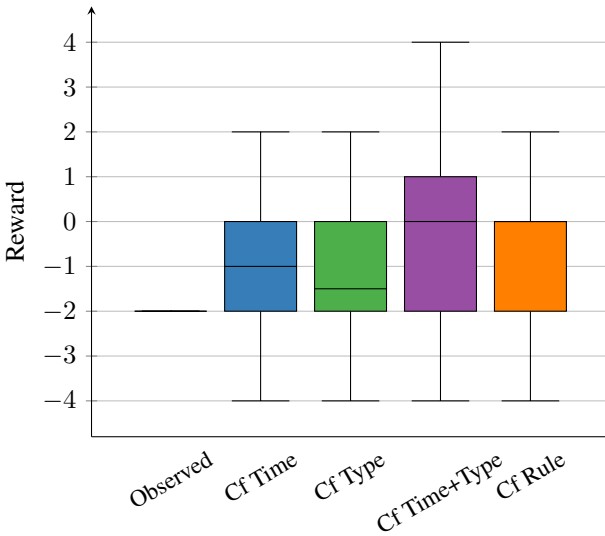

*Figure 2.* Comparison of Reward between Observed and Counterfactual Sequences

We considered seven treatment options, including Epinephrine (E), Norepinephrine (NE), Phenylephrine (PE), Epinephrine+Norepinephrine (E+NE), Epinephrine+Phenylephrine (E+PE), Phenylephrine+Phenylephrine (NE+PE), and Epinephrine+Norepinephrine+Phenylephrine (E+NE+PE). We record the last state of four lab measurements to calculate patients' survival time. Suppose each "normal" measurement contributes $+1$ to the survival time while "low" and "high" both contribute $-1$. If any lab measurement is unobserved, the contribution is set to $0$. The range of contribution of four measurements to survival time is $[-4, 4]$.

We estimated conditional intensity function $\lambda(s, t|H)$ using the observed patients data in MIMIC-IV, which models how the lab measurements are influenced by historical data and treatments. Then we counterfactual and optimize the treatment policies in both the timing and types.

**Results**  Fig.2 summarizes the observed sequence and counterfactual sequences for a patient whose initial reward is -2. In total, We finish four experiments: change the action time, change the action type, jointly change the time and type, and jointly change the time and type while considering the temporal logic rule constraint. For each experiment, we sample 25 counterfactual realizations to estimate the final reward. The mean reward value of four experiments are -0.87,-0.68,-0.24 and -0.44. This result shows that all counterfactual optimization sequences perform better than the observed sequence and jointly changing the action time and type performs best.

For the final experiment with temporal logic rule constraint, we adopt the rule to be "Epinephrine, Norepinephrine and Phenylephrine can't be used simultaneously".

To compare observed and counterfactual trajectories, we pick up a counterfactual sequence with reward 2 from the experiment of jointly changing action time and action type. Then we compare it with the observed trajectory in both action and progression of four states during the process. The result in Fig.3 shows that counterfactual trajectory has more normal states after treatment and thus has a higher survival time.

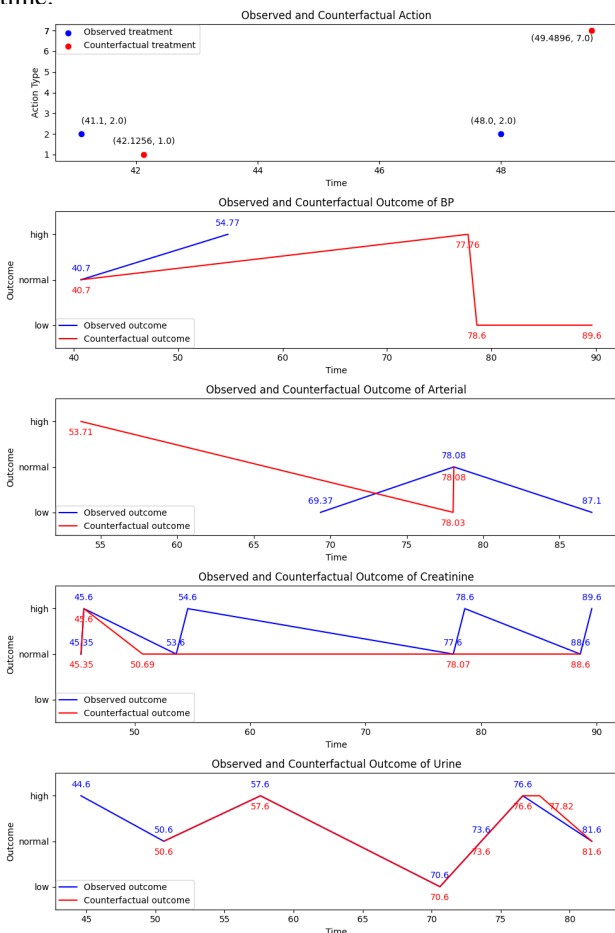

*Figure 3.* Comparison of States and Action between Observed and Counterfactual Sequences

# 7. Conclusion

In this paper, we propose a counterfactual optimization algorithm to answer what-if question: given the observational treatment and outcome trajectories, how could we perturb specific treatment actions to maximize the counterfactual outcome. We also apply temporal logic constraints to rule out rule out unreasonable counterfactual sequences. We evaluated our method on real MIMIC dataset and achieve promising results.

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

## A. Ogata Thinning algorithm

**Algorithm 2** Modified Ogata's Thinning Algorithm of MTPP

**Input** $: T, \lambda(t, s_k)(k = 1, 2...M), l(t)$
**Initialize :** $t = 0, H = \emptyset$
23 **while** $t < T$ **do**
24 $\quad$ $\lambda_{max}(t) = \max\limits_{t' \in (t, t+l(t))} (\sum_{k=1}^{M} \lambda(t', s_k))$
25 $\quad$ $\Delta t = \frac{-ln u_0}{\lambda_{max}}, u_0 \sim U(0, 1)$
26 $\quad$ **if** $\Delta t < l(t)$ **then**
27 $\quad\quad$ **if** $u_a < \frac{\sum_{k=1}^{M} \lambda(t+\Delta t, s_k)}{\lambda_{max}}, u_a \sim U(0, 1)$ **then**
28 $\quad\quad\quad$ Draw event type $s_k \sim \frac{\lambda(t+\Delta t, s_k)}{\sum_{k=1}^{M} \lambda(t+\Delta t, s_k)}$
29 $\quad\quad\quad$ $H = H \cup (t + \Delta t, s_k)$
30 $\quad\quad$ **end**
31 $\quad\quad$ $t = t + \Delta t$
32 $\quad$ **end**
33 $\quad$ **else**
34 $\quad\quad$ $t = t + l(t)$
35 $\quad$ **end**
36 **end**
37 **return** $H$

## B. Gumbel max Trick

### B.1. Draw samples from given category distribution

If the probability for discrete random variables $X_1, X_2...X_N$ are $\alpha_1, \alpha_2...\alpha_N$, we can use the softmax function to define the sampling probability $\pi_i$ of $X_i$:

$$\pi_i = \frac{\exp\{\alpha_k\}}{\sum_{k=1}^{K} \exp\{\alpha_k\}} \tag{7}$$

Meanwhile, we can also use Gumbel trick to achieve the same result, which is equivalent to adding the standard gumbel noise $g_k$ to the log-likelihood and take argmax of it. The distribution is the same as using softmax function

$$\arg\max_{k \in 1,...,N}(\alpha_k + g_k) \sim \frac{\exp\{\alpha_k\}}{\sum_{k=1}^{K} \exp\{\alpha_k\}} \tag{8}$$

$$g_k \sim \text{Gumbel}(0, 1) \tag{9}$$

### B.2. Get Gumbel distribution from given samples

Suppose a variable $X$ has a categorical distribution and we already observe the outcome $X_k$, we can also recover the posterior Gumbels noise that produces the result.

The maximum value is distributed as a standard Gumbel

$$\max_{k \in 1,...,N}(\alpha_k + g_k) \sim \text{Gumbel}(\log \sum_i \exp\{\alpha_k\}) \tag{10}$$

And the conditional probability of $p(g_i|K, g_k)$ is:

$$p(g_i|K, g_k) = \frac{f_{\log \alpha i}(g_i)[g_K \geq g_i]}{F_{\log \alpha i}(g_K)} \tag{11}$$

Note: [A] is the Iverson bracket notation: [A] = 1 if A is True, otherwise [A]=0

So for the Gumbel variable $g_k$ (where $X_k$ is the chosen variable) is

$$g_k = \log(\textstyle\sum_i \alpha_i) - \log(\alpha_k) \tag{12}$$

And the gumbel variable $g_i$ (where $X_i$ is not chosen) is

$$g_i = -\log(\exp(-\text{gumbel}(0,1) - \log(\alpha_i)) + \exp(-\text{gumbel}(0,1) - \log(\textstyle\sum_i \alpha_i)) \tag{13}$$

### B.3. Gumber-max Sampling Algorithm

Based on B.1 and B.2, we can sample posterior gumbel noise with following algorithm:

---
**Algorithm 3** Gumbelmax_sampling
---
**Input** : $\lambda_{obs}(t), \lambda_{cf}(t), \lambda_{max}, t, H$
**Initialize :** $G \sim Gumbel(0,1), \alpha_1 = \frac{\lambda_{obs}(t)}{\lambda_{max}},$
$\qquad\qquad \alpha_0 = 1 - \alpha_1, \alpha'_1 = \frac{\lambda_{cf}(t)}{\lambda_{max}}, \alpha'_0 = 1 - \alpha_0$

38 **if** $t \in H$ **then**
39 $\quad U_0 = \text{TruncatedGumbel}(\log(\alpha_0), G) - \log(\alpha_0)$
40 $\quad U_1 = G - log(\alpha_1)$
41 **end**
42 **else**
43 $\quad U_0 = G - log(\alpha_0)$
44 $\quad U_1 = \text{TruncatedGumbel}(\log(\alpha_1), G) - \log(\alpha_1)$
45 **end**
46 $X = \underset{i=0,1}{\text{argmax}}(\alpha'_i + U_i)$
47 **return** $X$

---

In the above algorithm, the truncated Gumbel is defined as

$$\text{TruncatedGumbel}(\log(\alpha_i), G) = -\log(\exp(-G - \log(\alpha_i)) + \exp(-G - \log(\sum_i \alpha_i)) \tag{14}$$

Here we only has two types: accept or reject and their sum $\sum_i \alpha_i$ is equal to one.

Belows is the counterfactual sample for marks (categorical distribution)

---
**Algorithm 4** Counterfactualmark_sampling
---
**Input** : $\lambda_{obs(t_0, s_i)}, \lambda_{cf}(t_0, s_i), s_0$
**Initialize :** $G \sim Gumbel(0,1), \alpha_1 = \frac{\lambda_{obs}(t)}{\lambda_{max}},$
$\qquad\qquad \alpha_0 = 1 - \alpha_1, \alpha'_1 = \frac{\lambda_{cf}(t)}{\lambda_{max}}, \alpha'_0 = 1 - \alpha_0$

48 **if** $t \in H$ **then**
49 $\quad U_0 = \text{TruncatedGumbel}(\log(\alpha_0), G) - \log(\alpha_0)$
50 $\quad U_1 = G - log(\alpha_1)$
51 **end**
52 **else**
53 $\quad U_0 = G - log(\alpha_0)$
54 $\quad U_1 = \text{TruncatedGumbel}(\log(\alpha_1), G) - \log(\alpha_1)$
55 **end**
56 $X = \underset{i=0,1}{\text{argmax}}(\alpha'_i + U_i)$
57 **return** $X$

---

## C. Counterfactual Optimization

Belows is the reinforce algorithm that calculate reward of counterfactual sequence $H'$ and optimize the policy $\pi_\theta$. Here we jointly change action time and type with policy $\pi_{\theta_t}$ and $\pi_{\theta_a}$. For each iteration, we sample n counterfactual trajectories to estimate the empirical value of gradient $\theta$. Each trajectory can be sampled by algorithm 1. We then calculate survival time and generate rewards to update our policy.

---

**Algorithm 5** Counterfactual Optimization

---

**Input** : $T$, Observed Sequence $H = \{(t_1, s_1), ..., (t_n, s_n)\}$, Dynamic model Z, Survival Function $J(t)$, Temporal logic constraints $L(a'_{1:n}) = \sum_{k=1}^{K} \omega_k \phi_k(a'_{1:n})$, Observed survival time $Y_0$

**Initialize** : Counterfactual Policy $\pi_\theta = \{\pi_{\theta_t}, \pi_{\theta_a}\}$, $\lambda_{obs}(t, s) = \lambda_{cf}(t, s)$ = Dynamic model(H)

58 **for** iteration *i=1,2...T* **do**

59   **for** counterfactual sequence $\tau = 1, 2, ..., n$ **do**

60     Counterfactual Sequence $H'_\tau$ = counterfacutal sampling $(T, \lambda_{obs}(t, s), \lambda_{cf}(t, s), H, Z, \pi(\theta))$

      Survival time $Y_\tau = J(H'_\tau)$

      Filter all counterfactual action $a'_{1:n}$ from $H'_\tau$

      Reward $G_\tau = (Y_\tau - Y_0) + L(a'_{1:n})$

      $\nabla\theta_{\tau,s} = \alpha\gamma^t G_\tau \nabla_\theta \log\pi_{\theta_s}$

61   **end**

62   $\theta_t = \theta_t + \epsilon\frac{1}{n}\sum_{\tau=1}^{n}\nabla\theta_{\tau,t}$

63   $\theta_a = \theta_s + \epsilon\frac{1}{n}\sum_{\tau=1}^{n}\nabla\theta_{\tau,a}$

64 **end**

65 **return** $\theta_t, \theta_a$

---

