# OpenReview forum: "Counterfactual Optimization of Treatment Policies Based on Temporal Point Process"
_ICML.cc/2023/Workshop/IMLH — IMLH 2023 PosterShortPaper_

### Official Review · Reviewer_FHyH · 2023-06-02

**Rating:** 7
**Confidence:** 3

**Review:**

The paper proposes a novel method of Counterfactual trajectories based on temporal point processes.
The paper is both well written and interesting. The inclusion of the predicate logic regularisation to control the potential counterfactual action recommendations is a quite nice addition. Sections 3 and 4  could be expanded a bit more for clarity and the specifics of the proposed algorithm. Line 73 column b is missing an "n" in the word "can".

Overall i believe this paper would be a welcome addition to the workshop

---

### Official Review · Reviewer_pU1L · 2023-06-17

**Rating:** 6
**Confidence:** 5

**Review:**

The authors present an early analysis of their approach to generate counterfactual treatment focusing on specific treatments.
- The paper is well presented and some efforts have been made to motivate the setup
- The Counterfactual Optimization has been well presented
- Initial results are promising

However, more efforts are needed to for a full publication. Arguments for point specific optimizations needs better justification and empirically motivated.

---

### Meta-Review · Area_Chair_1nZc · 2023-06-19

**Recommendation:** Accept (Poster)
**Confidence:** 4

**Metareview:**

The paper receive positive reviews on its counterfactual treatment optimization framework. I recommend acceptance.

---

### Decision · Program_Chairs · 2023-06-20

Accept (Poster Short Paper)